# Physical Activity Monitors in Companion Animal Chronic Pain Research—A Review Focused on Osteoarthritis Pain

**DOI:** 10.3390/ani15142025

**Published:** 2025-07-10

**Authors:** Connor Thonen-Fleck, Kate P. Sharon, Masataka Enomoto, Max LeBouef, David L. Roberts, Margaret E. Gruen, B. Duncan X. Lascelles

**Affiliations:** 1Translational Research in Pain, Department of Clinical Sciences, College of Veterinary Medicine, North Carolina State University, Raleigh, NC 27606, USA; cjthonen@ncsu.edu (C.T.-F.); menomot@ncsu.edu (M.E.); maxlebouef@gmail.com (M.L.); 2Elanco Animal Health, Greenfield, IN 46140, USA; kate.sharon@elancoah.com; 3Department of Computer Science, North Carolina State University, Raleigh, NC 27695, USA; dlrober4@ncsu.edu; 4Comparative Behavioral Research and Thinking Pets Program, Department of Clinical Sciences, College of Veterinary Medicine, North Carolina State University, Raleigh, NC 27606, USA; 5Department of Clinical Sciences, Comparative Pain Research and Education Centre, College of Veterinary Medicine, North Carolina State University, Raleigh, NC 27606, USA; 6Department of Anesthesiology, Center for Translational Pain Research, Duke University, Durham, NC 27710, USA; 7Thurston Arthritis Center, University of North Carolina, Chapel Hill, NC 27599, USA

**Keywords:** accelerometer, activity monitor, mobility, movement, wearable, dog, cat, pattern, pain

## Abstract

Movement and mobility are critical components of living, and so quality of life, for all animals, including pet dogs and cats. Many chronic pain conditions affect dogs and cats, and this pain can negatively affect movement and activities. Physical activity monitors (PAMs) are devices that can capture objective data on movement and thus have the potential to tell us about the impact of pain on movement and mobility and about strategies employed to reduce pain. However, in order to optimally use PAMs, researchers need to understand both device technology and biological influences. This review discusses the technical and biological considerations when applying PAMs to companion animal chronic pain research, in particular osteoarthritis pain research. It also provides an overview of use of these devices in veterinary chronic pain research thus far, and the potential of these devices in future studies.

## 1. Introduction

An organism’s interaction with its environment is the science of behavior, and such interaction involves movement. Movement can be affected by a variety of factors, one of which is pain. The experience of chronic pain is a complex and multi-dimensional physiological, functional, psychological, and social phenomenon in humans and is increasingly being recognized as such in companion animal species such as dogs and cats. Identifying and managing chronic pain is a field of growing interest in veterinary clinical sciences, but despite significant progress in the recognition and measurement of chronic pain, there is still an overall lack of relevant and validated methods for measuring the impact of chronic pain [1]. To further complicate matters, there is no one single outcome measure that quantifies changes in all the dimensions that can be impacted by pain; rather, different measures must be employed to understand the impact of chronic pain on an individual patient. Function and movement are impacted by pain, and include the amount, patterns, and quality of activity and mobility. These dynamic components of behavior can be measured by electronic devices, providing the opportunity to generate objective data about the impact of pain on movement behavior.

Adequate mobility is defined as the ability to move in one’s environment with ease and without restriction [2]. Increasingly, mobility is considered a vital component of health and quality of life in humans and a useful outcome measure [3,4]; similarly, the importance of mobility is being recognized in companion animals [5,6,7,8,9,10,11,12,13,14,15]. Pain impacts movement and mobility, and so the ability to reliably measure aspects of mobility has the potential to serve as a relevant endpoint in companion animal chronic pain research.

Movement can be tracked using wearable sensor-based technology as well as remote camera and other sensor technology in the environment. Recently, wearable technology such as accelerometer-based physical activity monitors (PAMs) have become increasingly popular, and are prime candidates for providing objective, non-invasive efficient systems for measuring free-living, habitual activity in companion animals. Although a number of PAMs exist on the market, to date, only a few devices have undergone direct assessment of validity, but in general they have been found to be valid surrogate measures of one or more aspects of spontaneous activity.

The ability to objectively measure and monitor activity could improve both diagnosis and the ability to accurately assess therapeutic efficacy in chronic pain conditions. However, variations in hardware, software, and data output and processing influence the function, reporting, and ultimately the interpretations of PAMs. Furthermore, there are multiple biological considerations when monitoring companion animal activity, such as age, body morphometrics, and species. Additionally, the way data are analyzed can profoundly influence inferences about activity. A general understanding of PAM technology and function as well as the influence of biological factors and approaches to analysis are important if the field is to advance.

The objectives of this review are to provide an overview of PAM technology, focusing on accelerometers, and describe technical and biological factors that influence PAM usage in companion animal chronic pain research. The most common form of persistent (chronic) pain in companion animals is osteoarthritis-associated pain, and most research on the impact of chronic pain has been performed on osteoarthritis pain. Hence, this review focuses on osteoarthritis pain, but the comments are likely relevant to all chronic pain conditions that may impact movement. This review also aims to provide an overview of applications of these devices in veterinary chronic pain research thus far, and the potential of these devices in future studies.

## 2. Materials and Methods

This systematic and commentary review was performed by searching Google Scholar and PubMed databases with combinations of the following keywords: physical activity monitor, activity monitor, chronic pain, veterinary medicine, osteoarthritis, and accelerometer. For articles generated from this search, the abstracts were read, and those relevant were downloaded and read in full. References cited within relevant papers were checked for any other reports that were relevant. These search results were combined with the authors’ expertise and opinions.

### Terminology

Accelerometer: A type of physical activity monitor (PAM) that utilizes accelerations generated by motion to measure activity—or strictly ‘changes in acceleration’—and therefore data which can be used to measure events such as whole body motion but also foot strikes and other events that result in changes in acceleration. This is in contrast to other types of physical activity monitors, such as gyroscopes, inclinometers, and pedometers, though a device may integrate multiple sensors at once to describe activity (e.g., inertial measurement units [IMUs]). Unless otherwise specified, PAMs referenced in this review refer to pure accelerometer PAMs.

Raw data: readings detected from accelerations at each sampling interval prior to any further data processing such as summation, rectification, etc.

Sampling interval: the number of times, measured in hertz, per second the sensor detects an acceleration reading. This rate determines the amount of movement detail over time that is being captured and potentially available for analysis if access to raw data is available.

Activity count: the final output from accelerometers. These are generic values representing average or total activity over a certain period of time, usually predefined by the user. These have undergone processing as dictated by the individual PAM device, which is typically a proprietary formula. There is no recognized ‘unit of activity’.

Epoch: user-specified time interval for which recordings are obtained. Epoch lengths determine the final output. An epoch represents the accumulated activity over the selected time, such as 10 s, 60 s, or 1 h.

## 3. Part I: Technical Considerations When Using Accelerometers

### 3.1. Accelerometer Technology

#### 3.1.1. Accelerometer Function

Fundamentally, PAMs measure body movement via an accelerometer. An accelerometer consists of a seismic mass that is displaced by static (e.g., gravity) or dynamic (e.g., subject movement-related) forces with the change in displacement measured as velocity. This combination of displacement, velocity, and force produces an electrical signal that is functionally equivalent to acceleration according to Newton’s Second Law (force = mass × acceleration). The final output is related to movement intensity and can be used as a surrogate measure of body movement.

The exact methodology for recording the displacement force within an accelerometer depends on the internal technology, of which there are three common types: piezoelectric, piezoresistive, and differential capacitance [16]. Piezoelectric and capacitance sensors are the most used in biological settings; piezoresistive is more commonly used in mechanical engineering contexts.

#### 3.1.2. Piezoelectric Accelerometers

Piezoelectric accelerometers contain a piezoelectric element (quartz or ceramic crystals) and a seismic mass housed inside a protective casing. During acceleration, the seismic mass causes the piezoelectric element to undergo deformation generating a proportional charge (voltage). The voltage is collected by electrodes and transmitted to a data processing unit, or a signal conditioner, by conductors, and represents the magnitude of acceleration.

#### 3.1.3. Capacitance Accelerometers

Capacitance accelerometers are manufactured with micro-electro-mechanical systems (MEMS) fabrication technology and are often referred to as capacitive MEMS accelerometers. These devices detect acceleration by a moveable plate, or “finger”, placed between two fixed plates. Displacement of the moveable finger due to acceleration changes the distance between the plates, producing a voltage output that is proportional to acceleration and collected similarly to piezoelectric devices. In general, these devices are suited to measure a low-frequency and low-magnitude acceleration (such as animal movement) as well as static (gravitational) acceleration. Capacitance devices are often used in consumer electronics.

#### 3.1.4. Axes Information

Accelerometers can measure acceleration in one, two, or three axes, depending on the design. In biology, information about movement in multiple axes is of interest and, to achieve this, either several unidirectional accelerometer units must be mounted orthogonally to each other or omnidirectional piezoelectric sensors that are capable of detecting accelerations in multiple axes can be used. Omnidirectional sensors detect accelerations in all three planes (*x*, *y*, and *z*), and are often referred to as ‘tri-axial’ accelerometers. Sensitivity in each direction is dependent on design. Devices vary in whether they combine the contributions from all axes into one output, or provide individual *x*, *y*, and *z* outputs. For example, the tri-axial device, Actical, outputs a single, combined three-dimensional output while the Actigraph GT3X can provide output for each of the three planes (*x*, *y*, and *z*) or a combination of axes via a vector magnitude equation, (*x*2 + *y*2 + *z*2)^1/2^, producing the composite acceleration.

Static and dynamic accelerations are measured by either direct current (DC) or alternating current (AC) response accelerometers, respectively. Capacitive MEMS accelerometers are DC-coupled and therefore capable of measuring static forces. Piezoelectric accelerometers are AC-coupled and consequently only capable of detecting dynamic events. An inability to identify the static state of gravity prevents such monitors from detecting angles which can be used to establish body position such as sitting versus lying down. Recent advances in solid-state technology, digital filters, and more complex analysis techniques have allowed the measurements of gravitational acceleration and therefore, body position [17,18,19].

### 3.2. Data Acquisition and Processing

#### 3.2.1. Data Acquisition

Data acquisition is determined by sampling frequency and measured in hertz (Hz), or the number of recorded values per second. For example, in a unit set to 50 Hz, a device will capture the acceleration 50 times per second. There is potential to miss certain movements if sampling frequencies are not appropriate for the targeted activity. The Nyquist stability criterion stipulates that the sampling frequency must be at least twice the frequency of the highest frequency movement [20]. For example, in dogs, running frequency is around 2 to 6 Hz [21], and to capture this, an acquisition frequency setting twice the running frequency would be needed. However, specific movements such as jumping or scratching are likely to be a much higher frequency. In order to collect data on the desired activity, the sampling frequency must be set to an appropriate rate as determined by the activity of interest. Often, the sampling frequency of a device is set, or limited options are provided for altering this.

#### 3.2.2. Data Filtration

Once sampled, data are filtered by a process called signal processing. Filtering the data removes or enhances specific frequencies in order to ‘clean’ up the data or remove signals or noise not of interest, like gravity, and allow for better detection of specific movements, such as steps [22]. Most filtering ranges in PAMs are between 0.25 and 7 Hz. Data filtration varies among individual PAMs and is one reason for differing output of different PAMs.

#### 3.2.3. Data Processing, Output, and Resolution

Once detected and filtered, the voltage signal undergoes analog to digital conversion, and then digital data sets undergo different device-specific processing techniques to reach so-called “activity counts”, or final outputs [23]. This can include rectification (the absolute value of accelerations is taken, eliminating the negative values caused by deacceleration), summation, area under the curve (a value is generated for each period of time, representing the intensity and duration of acceleration), or the change in activity from a running baseline threshold. Another approach is to determine the maximum value during an epoch which is then represented as the activity count for that time. In the case of tri-axial accelerometers, this value is often selected from the axis with the highest value. A mixture of these processing techniques, the exact combination of which is device-dependent and typically proprietary, yields a value for every sampling interval. Because of this, it is not meaningful to compare counts from different devices without a correction equation and, importantly, there is no recognized unit of ‘an activity count’. Activity counts are very much device-specific.

After these processing techniques, a PAM can produce summarized data based on the ‘epoch setting’. An epoch is a user-specified time interval over which recordings are reported, representing the accumulated activity over the selected time, which, depending on the selected PAM, can range from 1 s to 1 h. For example, if the epoch is set to one-minute intervals and the monitor sampling frequency is 32 Hz, there will be 1920 raw values processed internally for a final singular epoch-level observation (i.e., “activity count”). Most commercially available monitors do not offer access to raw values pre-summarization, which means the epoch selected defines the granularity of the data ultimately available. The balance between acquisition rate and epoch length is important to consider in relation to what the investigators wish to measure.

#### 3.2.4. Data Storage and Presentation

Device storage capacity varies, and time until the respective capacity is reached is dependent on the storage capacity and the selected epoch length (i.e., shorter epochs result in more data values accumulated in a fixed period of time). Final processed data are recorded onto internal memory and subsequently downloaded to a computer or transferred to cloud storage systems via Wi-Fi or via a mobile phone and Bluetooth. Several PAMs have associated software that is used to present the data on an interface. For a few monitors, access to the raw data (albeit with the effects of processing described above) is possible, but with most commercially available PAMs, presented data have undergone additional processing that is usually considered proprietary, making it difficult to interpret the output. PAMs currently (as of January 2025) utilized in, or marketed for, companion animals can be found in Appendix A. All the manufacturers were contacted at least twice requesting details to populate the table. The amount of information available on how the PAMs function is highly variable, and access to raw data is only possible for a few monitors. While this does not negate the value of a PAM to engage owners in activities with their pets, researchers ideally need access to raw data and a thorough understanding of how data have been processed and presented.

See Separate excel file in the Appendix A.

### 3.3. Device Calibration, Reliability and Validity

#### 3.3.1. Device Calibration

Although devices are calibrated during production, it is unclear if calibration is lost over time and, if this does occur, whether this influences accuracy [24]. Calibration of individual units is important with older sensors that employ a cantilever beam sensor with analog filtering, like older models of the Actigraph. Conversely, monitors utilizing direct compression technology and sensors with digital filtering, like newer models of the Actigraph, do not require unit calibration and an initial factory calibration is sufficient for the life of the device and have been shown to be accurate within 4% [24,25].

#### 3.3.2. Device Sensitivity and Validity

With any outcome tool or measure used in research, it is important to ensure that that tool measures what it intends to measure. Overall, this refers to the validity of that instrument.

The sensitivity of an outcome measure refers to that tool’s ability to detect changes, particularly clinically relevant changes. In the context of PAMs, this refers to the device’s ability to generate activity counts that change with relevant changes in physical activity.

There are several different layers to validity, including content (face), construct (convergent: how well the instrument agrees with other instruments measuring the same thing; and discriminant: how well the instrument can discriminate between the presence and absence of a condition), and criterion validity (how well the instrument agrees with the gold standard). Appendix A summarizes these measures in the context of PAM devices. As most accelerometer devices are created for use in humans, validation studies should be performed to verify that these devices do capture physical activity in companion animals, even though this assumption seems logical. An obvious example of why this is important is clear if one considers wrist-worn activity monitors such as the Apple watch—these use arm movement to measure activity (e.g., steps), but clearly this may not produce an appropriate measure of movement if mounted to the collar of a dog. A number of validation techniques have been used. These include checking videographic measures of motion against device activity counts, either by synchronizing filmed video with PAM output or by verifying counts to an ethogram [26,27,28,29] This type of validation asks the critical question, ‘does the PAM produce output that relates to movement?’ Another type of validation employs placing validated and new monitors on the same collar and comparing the output of the two (a form of construct validity) [30,31,32,33]. For example, Belda et al. found high correlation (r^2^ = 0.85) in total activity counts between the previously validated Actigraph GT3X and the PetPace monitor placed on the same collar [31]. However, this form of validation refers to convergent.

The environment in which a device is validated is also important. While studies performed in the laboratory are often the most convenient and an important step, the behavior of an animal within the laboratory versus its own home environment differs significantly [34,35]. Thus, validation studies in the laboratory may not accurately reflect the kinds of activity performed in the home environment.

It is also worth mentioning that validation studies are seeking to validate the accuracy of the final ‘activity count’ output as a measure of companion animal activity, rather than the accuracy of the device itself. As discussed above, it is important to remember that there is no such thing as a ‘unit of activity, or a standard ‘activity count’. Therefore, each PAM may represent the same biological activity with different ‘counts’ i.e., different device units are not equivalent. While manufacturers claim their device’s activity count output represents the biological activity of the subject, these data are not usually available to the public and such claims should be treated with caution.

#### 3.3.3. Device Reliability

Accelerometers have been reported to have considerable inter-device output variability between units of the same make, shown to be as high as 57% in Acticals [36]. This variability could be a result of several factors, including, but not limited to: calibration, age of the monitor, sampling frequency, or placement. Due to inter-device variability, selecting an analysis method that addresses this should be considered; using study designs that compare subjects (and therefore individual devices) to themselves may serve to address this issue.

#### 3.3.4. Data Summation and Analysis

There are several ways of examining (analyzing) the processed activity counts output from PAMs. The interval of summation or averaging will determine the level of detail that can be extracted from the data; for example, sums of activity over a day will conceal any hourly changes in activity. Similarly, alternative data reduction techniques result in different insights into the data. For example, summation of minute counts over an hour is representative of total activity over the hour, while a per-minute average over an hour cannot be thought of as the same thing. An appropriate summary interval function should consider what endpoint is being analyzed, and collapsing a time series may eliminate the details required for some interpretations of the data.

The majority of companion animal applications have traditionally focused on overall activity reflected by accelerometer activity counts, usually in the form of some type of regression analysis [26]. In most clinical studies, activity is reported using means, medians, and sums from a set time period (minutes, hours, days). For example, reported parameters used statistically have included: total activity count, [13,37,38] mean of activity counts, [12,13,32,37,39,40,41] median of activity counts [5,6,37,42] and percentage of change in activity counts [5,6,14]—all over varying time periods. Some studies have also examined activity intensity, however, a limitation of this is that the cut points between activity levels are often defined arbitrarily, either by the device or analyst; there is no standardized approach to analyzing or reporting intensities [7,13,43]. Other approaches to data analysis utilize subsets of the data, such as time of day, day of the week, or select time periods during the day [44]. Typically, traditional statistical approaches like the ones described above are limited in their ability to capture complex and nonlinear relationships in data because using averages of periods of time may obscure changes of interest. This was elegantly illustrated when comparing activity between healthy (non-painful) cats and cats with painful degenerative joint disease [45]. Healthy (non-painful) cats and cats with painful degenerative joint disease were no different based on mean per minute activity but differed in patterns of activity. More sophisticated analytic techniques designed to evaluate activity patterns are being increasingly used in veterinary chronic pain research, such as functional linear modeling (FLM) (see later) and functional data analysis (FDA) [46]. One issue in analysis of high-frequency longitudinal data is the high percentage of ‘zeros’, periods of no activity. Periods of no activity, such as rest, are the most frequent data point in studies of long-term habitual activity. One approach, hidden semi-Markov, is a statistical framework that prevents metric bias due to excess zeros, and this approach was successfully developed and applied to feline activity data to evaluate the effect of an analgesic treatment in cats with painful degenerative joint disease (DJD) [47].

Recently, various forms of artificial intelligence (AI), particularly machine learning (ML), have been applied to activity data, but this is still limited in veterinary chronic pain research [48,49].

### 3.4. Part I Discussion

Clearly, it is important to understand how PAMs function because variation in function creates opportunities for, or places constraints on, how well the output that is collected can be interpreted to measure the activity of interest. Critical features to understand are sampling frequency and epoch settings to ensure the activities of interest can be captured. The vast majority of PAMs marketed to the veterinary profession are ‘black boxes’ when it comes to understanding the way data are managed both on the device, and prior to presentation on an app interface. While proprietary claims about the interpretation of the data may be valid, this may not be appropriate for the research setting. Long epochs and post-collection ‘bucketing’ of data may obscure variations in activity that may be biologically important.

As more is understood about the complexity of activity in companion animals, and various methodologies are applied to increasingly granular data, it is critical to also understand factors and variability that affect the data in order to appropriately interpret the data.

## 4. Part II: Biological and Analysis Considerations

### 4.1. Biological and Use Considerations

Practical aspects such as placement of the monitor and attachment method as well as subject characteristics such as body conformation, weight, and age have the potential to influence physical activity data. Understanding these influences is important in designing studies and interpreting PAM output.

#### 4.1.1. Sensor Placement and Attachments

Variations in where accelerometers are placed on the body, use of an external case, the orientation of the unit, and the method of attachment to the body can all affect PAM output. In most clinical companion animal studies, accelerometers are placed on the collar [5,6,7,27,28,41,45]. In dogs, placement on the ventral portion of the cervical region has been suggested to yield reliable results, along with being convenient and most tolerable [26,37]. No effect on variability of total activity counts related to collar tightness was identified, although high inter-device variability was observed [36]. There has been some work evaluating the impact of device location on an animal’s body and the subsequent effects on variability and validity of accelerometers relating to correlations with distance moved [26]. Hansen and colleagues reported that all mounting locations examined (top of a collar, bottom of a collar, lateral portion of the thorax in a vest, axilla region, lateral portion of the humerus, antebrachium, under the sternum, and under the abdomen) provided acceptable correlation with videographic measurements of movement and mobility, and the ventral portion of the collar was determined to be the most convenient location. Correlation between movement and mobility (distance traveled, time spent walking, and time changing position) and 1 min total activity counts were similar among all locations (*R*^2^ = 0.71–0.93). Importantly, the widely used and convenient location—the ventral portion of the collar—was highly correlated (*R*^2^ = 0.89) with distance travelled. Accelerometer placement (on a collar with a pouch, on collar without a pouch, on a harness with a pouch, and on a harness without a pouch) and the ability to differentiate between rest and movement as well as increases in gait speed, was investigated in dogs [50]. Only pouched devices on a harness could detect walking speed increases from 5 km/h to 7 km/h on a treadmill, and inclination of a treadmill (5%) did not alter activity count output. The attachment of leashes to collars is another complicating factor of PAM usage. In dogs, leash attachment to the collar with the PAM mounted on it led to poor correlations between physical activity monitors, indicating that leash attachment to the PAM collar influenced data acquisition [51]. It is recommended that a second, dedicated collar is used with PAMs to avoid leash interference.

In contrast to the overall findings in dogs, in cats, monitor placement has been shown to influence activity counts [27]. Higher activity counts were reported from collar compared to harness-placed Actical monitors, using video-measured distance moved. This may have been partly due to the fact that collar-mounted monitors produced activity counts during eating and grooming activities. However, overall, when compared with distance moved, correlation was excellent between collar and harness mounted accelerometers, *r* = 0.80 and 0.90, respectively.

#### 4.1.2. Body Conformation

Overall, body size and conformation affect how animals move; gaits differ and therefore movement frequencies differ and acceleration is expected to be dissimilar. However, Brown and colleagues [37] found that no aspects of body conformation (including weight, body condition score, circumference, and body length) significantly affected total activity counts when dogs were walked and trotted in a controlled environment, but the authors did suggest body condition score and body length be considered in multivariate models. However, total activity counts when traveling up and down stairs were influenced by body conformation, specifically body weight and age, and when controlling for age, every 1 kg increase in body weight was associated with a 1.7% decrease in activity counts [37]. Overall, the study concluded that when activity was less controlled, older dogs and larger dogs had lower activity counts than younger and smaller dogs. Similarly, in another study on acute pain, the authors found activity varied with body weight, with smaller dogs showing greater decreases in activity after surgery [9]. In dogs with osteoarthritis pain being enrolled into a study, Labrador retrievers had significantly lower activity counts when compared to non-Labrador breeds at the time of enrollment [52], but whether this is conformation-related or breed-related per se, is unknown. Labrador retrievers also showed a larger increase in higher-intensity activity when administered an NSAID for pain relief compared to other breeds. It has also been found that morphometrics, particularly shoulder height, significantly relate to PAM output and distance traveled [22]. Clearly, even within a relatively narrow range of sizes of dogs, body conformation has effects on activity output and should be considered when interpreting data when between-subject comparisons are made. No studies have been performed in cats to study the effect of breed or bodyweight on PAM-measured activity.

#### 4.1.3. Age

Young dogs have been shown to be more active (rearing, jumping, grooming, and distance moved) than mature dogs in a laboratory setting [35,53]. Even within a fairly narrow range of ages of pet dogs (mean of 4.9 ± SD of 3.2 years), a 1-year increase in age resulted in a 4.2% decrease in total activity counts in dogs measured with an Actical device set at a 1 min epoch while dogs performed set activities [37]. Some conditions, notably OA and DJD, are more common in older animals and, although pain associated with these conditions may impact the level of activity, it is clearly important to control for age when comparing healthy animals to those with chronic pain conditions. In general, there is a lack of understanding of the effects of age on activity, and there are no published data on ‘normal’ levels of activity across age ranges for any monitor device. It is also important to consider the influence of other age-associated conditions, such as age-related cognitive dysfunction. It has been shown that activity is impacted (including patterns, see later) in cognitively impaired dogs, with activity counts generally being higher in association with cognitive dysfunction (likely due to pacing) [35], similar to some human patients with cognitive dysfunction [54], but differing from humans with mild cognitive impairment where activity was lower [55].

Much less has been studied regarding the influence of age on activity in cats. Interestingly, when comparing activity in healthy cats and those with DJD-associated pain, Gruen and colleagues [45] found that the range of activity counts for normal cats was smaller than for cats with DJD, however the mean average per-minute activity for each group was not significantly different. This is of interest here because the healthy cats were significantly younger (mean 5.8 years) than the cats with DJD (mean ~ 12 years). In contrast to this suggestion of a minimal effect of age, a recent study [56] found that, in general, junior cats spent more time being active than both prime and mature cats in summer and winter, but there was an influence of season, with difference between younger and older cats accentuated by the winter season. These data bring up an important point which is often overlooked when reading literature on PAMs—different authors reporting different parameters that sound similar. Gruen et al. reported activity counts (mean per minute), while Smit et al. reported ‘time spent active’, and these are different measures.

### 4.2. Patterns of Activity

Overall, work with PAMs has shown that activity varies across a 24 h period, and this subsequently led to exploration of activity patterns and how they may differ with painful disease and treatment. Just as with aggregate summaries of activity data, understanding how patterns vary is a prerequisite to understanding how chronic pain affects these.

Activity in cats has been shown to be bimodal with higher activities recorded in the early morning and late at night both in laboratory-housed research cats [7,57] and client-owned cats in their home environment [45]. In early work looking at patterns (constructed using average hourly activity counts) 24 h patterns were shown to vary with exposure of cats to short (8 h light, 16 h dark) or long (16 h light, 8 h dark) day periods [57]. Similar work in client-owned pet dogs, using mean activity counts per minute for each hour, found a bimodal pattern of activity over the 24 h period [44]. In the client-owned settings, interactions between owners and their pets likely influence this pattern. This may be indirect such as activity initiated by owners waking up in the morning, or more direct such as taking pets for walks. Although this influence has not been studied in detail with carefully documented owner activity, the morning and afternoon peaks of activity shown by both dogs [44,58] and cats [45] coincide with times when households are generally active, pets are fed, and dogs are taken for walks. In cats, weekend activity patterns differed from weekday patterns [45], with the peaks and troughs of activity being less pronounced at the weekend and this is assumed to be due to different patterns of owner activity at the weekend compared to weekdays. The effect of human interactions has also been seen in laboratory animals—decreased presence in the facility resulted in lower total activity counts at the weekends in cats [27]. In healthy (non-painful) pet dogs [38] and dogs with OA pain [58] activity at the weekends has been found to be higher than during weekdays. Interestingly activity in healthy (without chronic pain) cats was found to be lower during weekends compared to weekdays [45].

The understanding of variation of activity over a 24 h period has led some investigators to focus on certain time segments to answer specific questions. For example, determining the effect of chronic pain on nighttime activity, or ‘restfulness’. The nighttime period has been arbitrarily defined (e.g., 00:00–05:00 h, or 22:00 and 06:00 h) or defined based on owner diaries of when the household went to bed and got up [59]. In two independent studies using different statistical approaches [60,61] provision of known analgesics (NSAIDs) resulted in a decrease in nighttime activity, suggesting that the OA-pain state results in increased nighttime activity, which could be restlessness as reported in humans with symptomatic OA [62]. Further, in humans, total hip replacement resulted in sleep becoming less fragmented as measured by accelerometry [63]. It is interesting to note that, in one study in dogs, when data were analyzed using summary statistical approaches (bucketing the data into a single value) [59] no effect of the NSAID was seen, and a decrease in activity only detected when a ‘pattern’ analysis (functional linear modeling, FLM) was used [60]. This shows the importance of considering the approach to analyzing the data (see later).

As alluded to earlier, analysis of patterns of activity appears to have potential for defining the effect of covariates on activity. Early work identified changes in activity over 24 h periods [35,44,58] but used summary values for each hour. Smoothing approaches offer a more nuanced evaluation of activity patterns. Gruen and colleagues used Functional Data Analysis (FDA) to compare patterns of activity in healthy cats and cats with painful DJD, and to compare patterns at the weekend with weekday [45]. FDA has also been applied to jumping activity in cats, using FDA to produce curvilinear data signatures in order to be able to analyze differences in different types of jumping [46]. A branch of FDA, functional linear modeling (FLM), has recently been applied to activity data. FLM is a computational approach for analyzing and visualizing high-frequency longitudinal data such as physical activity and can be used to assess the influence of individual factors on activity patterns [45,60,64,65,66]. By converting raw activity counts into a continuous curve over time and comparing these sets of functions, one can visualize significant effects of covariates (i.e., age, body weight, signalment) without sacrificing granular information typically lost with summary statistics. FLM corrects for weaknesses present in traditional modeling by allowing data to be ‘smoothed’ to detect variations in the pattern across a 24 h period [66]. As described above, Gruen and colleagues used FLM to evaluate the effect of analgesic provision on nighttime sleep [60]. More recently, FLM was used to evaluate the effect of age/fractional lifespan, joint pain, and cognitive function on activity patterns in aged dogs [65]. The investigators found all these factors influenced activity patterns, but importantly, the effects were not necessarily uniform across 24 h—for example, older age was associated with dramatic reductions in activity in the evening, but older age was associated with more activity during the middle of the day [65]. Similar findings were reported recently in a cohort (*n* = 99) of dogs with OA pain, with strong effects of age, and (within the cohort of dogs with OA pain), greater joint pain decreasing activity in the evening [67]. The investigators also found that hindlimb pain had a greater impact on activity patterns than forelimb pain [67]. If activity patterns are to be used to assess the impact of analgesic therapies, then it is important to know how phenotypic characteristics influence activity so that groups of differing phenotypes can be evaluated appropriately. The R packages used in the FLM analyses that have been reported can only evaluate one covariate at a time [45,60,64,65,66], but recent advances in functional data analysis allow for the influence of several covariates on activity patterns to be statistically evaluated [55].

Clearly, time of day, owner activity, characteristics of the animal and pain status all affect pet activity. Whether analyzing select periods of the day, or evaluating activity patterns, biological justification of the approach is important. More research is needed to understand the factors that influence activity and activity patterns in order to optimally use and interpret changes in activity in relation to pain states or analgesic provision.

### 4.3. Biologic Meaning of Changes in Activity

In addition to understanding what factors influence activity, we need to concurrently understand the clinical meaning of changes in activity. For example, after administration of an analgesic, the treated group may have a statistically significant increase in activity counts, but this leads to a very important question: ‘is this clinically important?’ Approaches to help define a clinically meaningful change include the use of effect size (ES) and number needed to treat (NNT). However, for the latter (NNT), a threshold for success is needed. With activity data, no research has focused on what a successful change would be—and indeed, any threshold would very much depend on how the data were managed and analyzed. One approach aiming to understand the clinical meaning of changes in activity is to generate data using known analgesics and then comparing these data to novel or putative analgesics. For example, in a placebo-controlled study of an anti-nerve growth factor monoclonal antibody, over the first 3 weeks following administration of frunevetmab a 13% increase between treatment and placebo groups’ change in activity from baseline was seen, with the treatment group showing an increase in activity from baseline [14]. This can be put into perspective using data from another study that evaluated a ‘known’ analgesic, the NSAID meloxicam [41]. In the latter study, the difference between placebo and treatment groups in change in activity over a similar period of time was 3.32%, suggesting the change in activity from the anti-NGF treated cats (13%) was potentially clinically meaningful—at least based on the assumption that NSAIDs enhance mobility in cats with painful joint disease. Much more research needs to be performed in this area to define what meaningful changes in measured activity are, and thresholds for success (using, for example, anchor-based methods [68].

### 4.4. Detecting and Understanding the Effects of Analgesics in Chronic Pain Conditions

It is intuitive that chronic pain decreases activity, however this may be a too simplistic view. As described earlier, in cats it has been reported that overall activity levels (mean per minute activity) were not different between healthy cats without joint pain, and those with clinically apparent joint pain from OA [41]. In dogs, one study has compared activity levels (average hourly activity counts) between healthy dogs and dogs with OA pain, and found activity was significantly lower in the OA pain dogs, although the OA pain dogs were significantly older (and increasing age decreases activity) [69]. When evaluating patterns of activity in dogs with OA pain, at times of peak activity within a 24 h period, dogs with higher owner-assessed pain/disability scores, higher joint pain, and greater muscle atrophy all had decreased activity profiles, suggesting chronic pain does decrease activity [67]. However, factors were evaluated individually for their effect on activity patterns, and not in a combined statistical model. Overall, it has become clear that the effects on activity levels are not as dramatic as might be assumed, with other factors (e.g., age, owner-instigated changes in activity) having potentially larger effects [70]. Despite this, in a study looking at how initial degree of impairment (based on owner assessments) influenced response to analgesic therapy (NSAID) in dogs with OA pain, Muller and colleagues found that greater initial impairment was associated with smaller mean weekly and weekday activity counts (averaged across dogs) [52], and larger positive changes in weekly and weekday activity expressed as a percentage change from baseline, compared to dogs with less initial impairment [52]. The approach of using percentage change, versus group means, was taken to try to account for the large intra-subject variation in activity. This has also been seen in cats [41] and the large intra-individual variation in activity in both dogs and cats suggests that evaluation of changes within individuals may be more relevant than looking at group effects.

Another assumption that has been made is that activity will remain relatively static in placebo-treated animals over time and increase in treated animals. However, in early work in cats, it was found that, although anti-NGF mAb treatment increased activity over baseline levels [14], activity in the placebo group gradually declined, even over the relatively short 9-week period of the study. A similar pattern of decreasing activity over time was seen in a follow-up study, where both the placebo and treated groups had an overall reduction in activity over the 8 weeks of the study, with the treated group showing significantly less reduction in activity [71]. Similarly, in dogs with OA pain, provision of an NSAID resulted in an initial increase in activity (as seen in the cats [71]), and then a gradual decrease over the 16 weeks of the study. In these examples, there appeared to be a benefit of analgesic provision, but clearly there is more to understand about ‘normal’ activity over time, especially in the context of clinical trials.

In cats, treatment with meloxicam, tramadol, robenacoxib, and anti-NGF mAb have all resulted in positive influences on activity [14,39,47,72]. On the other hand, treatment with gabapentin or amantadine resulted in decreased activity likely due to the sedative effects of the drugs, despite improvements in owner assessments of pain and function [73,74]. In contrast, in dogs with osteosarcoma pain, an NSAID improved activity (more ‘high’ activity and less ‘low’, or ‘restless’ activity as defined by activity counts), and the addition of gabapentin did not decrease the improvement in high activity [75].

Clearly, there is much to learn about how pain states influence activity, and how pain relief affects this to optimally use PAMs to assess the impact of analgesic treatment. In a thought-provoking study in mice, Bohic and colleagues applied supervised and unsupervised machine learning to 3D pose data (movement) and found that movement was robustly altered in each of the chronic pain models, but commonly used analgesics did not return the animals’ movement to pre-injury states, rather the analgesics induced a novel set of spontaneous behaviors not observed prior to induction of the pain model, or prior to treatment [76]. Perhaps a similar situation may occur in cats and dogs with chronic pain, so an expectation to ‘normalize’ activity is unreasonable.

### 4.5. Future Directions

#### 4.5.1. Integrated Units

Accelerometers detect linear accelerations in the *x*, *y*, and *z* planes. However, mobility is difficult to define simply in terms of linear motion. Furthermore, as described above, some type of force on the accelerometer is required to generate acceleration. Thus, behaviors without motion, like sleeping versus simply sitting still, are difficult to distinguish from each other as no acceleration is present [77]. To avoid these limitations, accelerometers can be combined with other sensors, like gyroscopes, magnetometers, inclinometers, and global positioning system (GPS), which measure angular velocity, changes in the magnetic field, more precise accelerations, and position, respectively [77,78,79]. Such combinations of sensors are termed ‘inertial measurement units’ (IMUs). These devices have the ability to collect more comprehensive information about an animal’s motion and have been utilized to study gait and posture, specific activities or behaviors, and the quality of motion and activity [79,80,81,82,83]. Despite the advantages of having data from multiple sensors, there are several disadvantages to an IMU-based PAM versus a pure accelerometer. IMUs have the disadvantage of a significantly shorter battery life, thus usually limiting their usage to short-duration (e.g., 2–4 h) collection of data in artificial research environments. Another disadvantage is angle calculation using magnetometers is not that accurate, especially in the frontal and transverse planes compared to motion capture.

#### 4.5.2. Detecting Specific Movements and Behaviors

Anyone who has suffered a chronic pain condition knows that activity and activities of daily living are not all affected in the same way—for a given pain condition, performing one activity may be difficult, whereas another activity may not be impacted. It follows that detecting specific movements and behaviors may be of utility in assessing analgesics. Of course, the success of this approach relies on knowing (1) the accurate phenotype of the animal being assessed (e.g., knowing and labeling a dog with hip and stifle OA pain, versus labeling the dog as ‘a dog with OA pain’) and (2) the specific movements and activities that are impaired by each pain condition or phenotype. Capturing information on specific activities requires a consideration of acquisition capability and epoch length for the PAM. For instance, identifying jumping in a cat (approximately 0.1–0.3 s) would require a high acquisition rate and access to the raw data. If high frequency data can be acquired, and transferred to storage, ML can be leveraged to develop algorithms to identify specific behaviors, as has been done for the identification of scratching in dogs [84]. Further, as each type of movement or activity is associated with a unique signature of change in acceleration (as has been demonstrated for different types of jumping in cats [46]), there is the potential to not only identify when an activity is performed, but how (normally) it is performed. Thus, the effect of analgesics on ‘normalizing’ how specific activities are performed could be used as an outcome measure. The caveat to this is that even in the absence of pain, other factors (e.g., joint pathology due to osteoarthritis) may not allow performance of an activity to be normalized.

ML-based behavior classification has been a powerful tool in several fields of study, but this technology has not yet been effectively translated to veterinary chronic pain research. In part, this can be attributed to the complexity of habitual activity; complex, long-term activities are an accumulation of simple activities contextualized by biological factors like time of day, presence of other species, and routine, thus, ML algorithms must be trained for a variety of factors outside of the activity count [85]. Furthermore, the sampling frequency required to capture specific movements, such as jumping, is extremely high and requires massive quantities of data to be collected, making training ML algorithms laborious.

Despite these barriers, the use of ML to detect specific behaviors of interest would be an incredibly useful tool in veterinary chronic pain research. Initial work has been done in dogs in controlled environments, where good accuracy at detecting several behavioral features utilizing machine learning was achieved via an accelerometer sensor attached to the ventral neck and back/harness [86,87]. It will be important to translate these methods from controlled environments to free-living animals for the full impact to be realized. As discussed above, accelerometers capture data based on motion and velocity; thus, ML algorithms applied to accelerometer data often perform poorly at determining static behaviors such as laying down, sitting, etc. IMUs have been used to bridge this gap, allowing for good estimation of posture in dogs [87]. This study required multiple IMU units mounted on the dogs, which is unlikely to be feasible in a clinical trial environment. However, in recent work, investigators were successful in detecting motion differences between healthy cats and cats with OA pain using an ML algorithm applied to activity data from a collar-mounted accelerometer [48]. They focused on activity just prior to jumping in the cats.

The ability to quantify and assess more specific behaviors offers the potential of objectively identifying early signs of behavioral changes that may be missed by owners and veterinarians. Algorithms like these, combined with instantaneous accelerometer data transfer, could be the key to non-invasively and objectively identifying early signs of chronic pain conditions like OA.

#### 4.5.3. Smoothness of Motion

Currently, the field of veterinary medicine has almost exclusively focused on total or mean activity in set periods of time, or patterns of activity over 24 h periods, and, more recently, on identifying specific behaviors of interest. Thus far, accelerometry has been used to provide a quantitative measure of movement, not a qualitative one. Pain may cause animals not to move less, but to move differently. This may be captured by an analysis of ‘how’ specific activities are performed (for example jumping in cats [48]), and, related to this, another approach might be to look at how ‘smoothly’ activities are performed. Movement smoothness is defined as the continuity or disruption of a specific movement, like taking a step or drawing a circle. Utilizing accelerometers and/or gyroscopes to measure disruption in movement is of particular interest because pain can cause a disruption of smoothness of movement [88,89]. Work needs to be done to ascertain whether smoothness of motion is altered in companion animals due to pain, whether it is measurable, and whether classic ‘jerk’ metrics are applicable [89,90] or other analytical approaches need to be employed.

#### 4.5.4. Implantable Accelerometers

In companion animals, it is already standard practice to use subcutaneously implanted microchips for pet identification. With technological advances, real time physiological monitoring, including activity, is possible using implanted microchips [91,92]. While battery life is currently limiting, so called biobatteries are on the horizon [93]. It is likely that in the future, we will utilize implantable sensors to gather physiological data, including activity, from animals in real-time. The challenge, similar to where we are at the moment, will be to understand the biological meaning of the activity data generated.

#### 4.5.5. Future Integration of PAM Data into Research

Readers of this review might be left with the impression that the use of PAMs in chronic pain research are an alternative measure of the same latent construct (pain). Indeed, this is how the field currently views such sensor data—an alternative dimension. We believe the future is to consider PAMs and other sensor data, as well as all other measures of the impact of pain, as components of the complex behavioral repertoire of the individual. Such a network approach to measuring the impact of pain will advance our ability to individualize measurement and therefore successful treatment.

### 4.6. Part II Discussion

In companion animal chronic pain research there are several factors that influence activity counts, which should be considered in designing and interpreting studies. These include monitor placement, body conformation, age, owner involvement and modulation of activity, and the presence of other conditions that may alter activity (e.g., cognitive impairment, cardio-vascular disease). Movement is complex and varies with time of day, presence of other animals, routine and human interaction. This, coupled with the fact that different types of activity may be differentially affected by pain suggest that analysis of patterns of activity, or even specific activities, may be most informative in the context of chronic pain research. Managing zero-inflated, high frequency, longitudinal data to evaluate patterns is becoming facilitated by greater understanding of and use of modern software, and advances in statistical approaches. However, ultimately, advances in the application of PAMs to chronic pain research will require a greater understanding of the biological meaning of activity and changes in activity.

## 5. Conclusions

We generally accept that chronic pain alters activity. PAMs allow for objective measurement of activity and can be applied to companion animal pain research. Although PAMs have shown promise in pain research, a greater understanding of PAM technology, biological factors influencing activity, how chronic pain changes activity, and of data management and statistical approaches to analyzing data, are all critical to advance the field. Importantly, across the many accelerometers being marketed to the animal health sector, there are few data validating the claims being made, and little transparency in the technology, making it difficult to independently advance the field. However, PAMs have the potential to reveal important information about the patterns and types of movement, how specific activities are performed, and the influence of pain on these aspects. Advances in this field have the potential to provide novel, clinically relevant, objective outcome measures for chronic pain research.

## Data Availability

Not applicable.

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
