# Peer review of "Physical Activity Monitors in Companion Animal Chronic Pain Research—A Review Focused on Osteoarthritis Pain"

_animals, 2025, doi:10.3390/ani15142025_

Round 1
Reviewer 1 Report
Comments and Suggestions for Authors
Overall, a well written and useful paper that adds to the literature.
65– reference to add to 5-14
Carr B, Levine D, Marcellin-Little D. Gait Changes Resulting from Orthopedic and Neurologic Problems in Companion Animals: A Review. Advances in Small Animal Care. 2023 June. doi: https://doi.org/10.1016/j.yasa.2023.05.001
102 – accelerators are also used for event detection such as foot strike in gait studies.
308 – reference to add on validity
Hyytiäinen H, Levine D, Marcellin-Little D. Clinical Instruments for the Evaluation of Orthopedic Problems in Dogs and Human Patients, a Review. Advances in Small Animal Care. 2023 July. doi: https://doi.org/10.1016/j.yasa.2023.05.007
596 – just say increase rather than delta
601 – is 3% meaningful? Maybe potentially meaningful?
621 – age and owner involvement? Likely the owner following a plan would increase activity more than anything. Morrison R, Reilly JJ, Penpraze V, Pendlebury E, Yam PS. A 6-month observational study of changes in objectively measured physical activity during weight loss in dogs. J Small Anim Pract. 2014;55(11):566-570. doi:10.1111/jsap.12273
679 – what is shorter? Maybe up to 6 hours or something like that. Another disadvantage is angle calculation using magnetometers is not that accurate, especially in the frontal and transverse planes compared to motion capture.
773 –owner involvement as previously stated
745 – angular velocity can be used to look at smoothness of motion or quality of motion – this is primarily the gyro data not the accelerometer data in the human field and is evolving in the companion animal field.
Author Response
Response to reviewers
The authors would like to thank the reviewers for their thoughtful and insightful comments.
Reviewer 1
Overall, a well written and useful paper that adds to the literature.
REPLY: Thank you for your review and comments – they have helped improve our submission!
65– reference to add to 5-14
Carr B, Levine D, Marcellin-Little D. Gait Changes Resulting from Orthopedic and Neurologic Problems in Companion Animals: A Review. Advances in Small Animal Care. 2023 June. doi: https://doi.org/10.1016/j.yasa.2023.05.001
REPLY: Thank you – I’d missed this. Great review.
102 – accelerators are also used for event detection such as foot strike in gait studies.
REPLY: Good point – that sentence has now been updated to: Accelerometer: A type of physical activity monitor (PAM) that utilizes accelerations generated by motion to measure activity – or strictly ‘changes in acceleration’, and therefore data can be used to measure events such as whole body motion but also foot strikes and other events that result in changes in acceleration.
308 – reference to add on validity
Hyytiäinen H, Levine D, Marcellin-Little D. Clinical Instruments for the Evaluation of Orthopedic Problems in Dogs and Human Patients, a Review. Advances in Small Animal Care. 2023 July. doi: https://doi.org/10.1016/j.yasa.2023.05.007
REPLY: Again, I’d missed this one – thank you
596 – just say increase rather than delta
REPLY: Changed
601 – is 3% meaningful? Maybe potentially meaningful?
REPLY: Wording has been altered to include ‘potentially’ and make it clear what study was being referred to
621 – age and owner involvement? Likely the owner following a plan would increase activity more than anything. Morrison R, Reilly JJ, Penpraze V, Pendlebury E, Yam PS. A 6-month observational study of changes in objectively measured physical activity during weight loss in dogs. J Small Anim Pract. 2014;55(11):566-570. doi:10.1111/jsap.12273
REPLY: Thank you – the potential effects of owner instigated activity has been added, and the reference suggested also added
679 – what is shorter? Maybe up to 6 hours or something like that. Another disadvantage is angle calculation using magnetometers is not that accurate, especially in the frontal and transverse planes compared to motion capture.
REPLY: Thank you – a time-frame is given (2-4 hours) to indicate what is considered short, and the additional disadvantage of IMUs has been added.
773 –owner involvement as previously stated
REPLY: Added
745 – angular velocity can be used to look at smoothness of motion or quality of motion – this is primarily the gyro data not the accelerometer data in the human field and is evolving in the companion animal field.
REPLY: Thank you – this has been added. Interestingly however, in our recent work, accelerometer data features (e.g. Jerk) were better able to distinguish OA pain from healthy control dogs, using CNN approaches. So much more to be learned!
Reviewer 2 Report
Comments and Suggestions for Authors
Pain recognition is an important topic in veterinary medicine. key factors for pain recognition are various and must be integrated within each other to correctly quantify the degree of pain.
Pain is related to reduced movement. the authors aim to provide a narrative review about the state of the art in the use of accelerometers in veterinary medicine, their technology and biological interpretation.
the paper is well written. as a narrative review, not much about methodology is described. However since an extensive literature analysis has been made, I would suggest the authors to consider changing the narrative into a systematic review.
I have a couple of comments:
- since in the keywords osteoarthritis is mentioned I would change the title accordingly (movement reduction is not related only to orthopedic pain....)
- I can't find table 3
- the supplementary material is not mentioned in the text (or at least I couldn't find it)
Author Response
Response to reviewers
The authors would like to thank the reviewers for their thoughtful and insightful comments.
Reviewer 2
Pain recognition is an important topic in veterinary medicine. key factors for pain recognition are various and must be integrated within each other to correctly quantify the degree of pain.
REPLY: Agreed – and another interest of ours is the integration of disparate sets of data collected using various measurement techniques in order to better classify the pain state.
Pain is related to reduced movement. the authors aim to provide a narrative review about the state of the art in the use of accelerometers in veterinary medicine, their technology and biological interpretation.
The paper is well written.Aas a narrative review, not much about methodology is described. However since an extensive literature analysis has been made, I would suggest the authors to consider changing the narrative into a systematic review.
REPLY: We have considered the suggestion and agree. Our approach was that of a systematic review, and so we have changed ‘narrative’ to ‘systematic’.
I have a couple of comments:
- since in the keywords osteoarthritis is mentioned I would change the title accordingly (movement reduction is not related only to orthopedic pain....)
REPLY: Thank you. Including the above comment, we have now changed the title to: Physical activity monitors in companion animal chronic pain research – a review focused on osteoarthritis pain.
Additionally, we have made it clear that we are focusing on osteoarthritis pain because that is where most of the scientific information is, but left comments to acknowledge that it is not only OA pain (and orthopedic conditions) that affect movement.
- I can't find table 3
REPLY: We have searched, and do not appear to have referenced a ‘table 3’.
- the supplementary material is not mentioned in the text (or at least I couldn't find it)
REPLY: The supplementary file is referred to in section 3.3.2